# Membrane Microvesicles as Potential Vaccine Candidates

**DOI:** 10.3390/ijms22031142

**Published:** 2021-01-24

**Authors:** Layaly Shkair, Ekaterina E. Garanina, Robert J. Stott, Toshana L. Foster, Albert A. Rizvanov, Svetlana F. Khaiboullina

**Affiliations:** 1Institute of Fundamental Medicine and Biology, Kazan Federal University, 420008 Kazan, Russia; shkair.layaly.94@gmail.com (L.S.); EEGaranina@kpfu.ru (E.E.G.); rizvanov@gmail.com (A.A.R.); 2M.M. Shemyakin-Yu.A. Ovchinnikov Institute of Bioorganic Chemistry of the Russian Academy of Sciences, 117997 Moscow, Russia; 3Faculty of Medicine and Health Sciences, School of Veterinary Medicine and Science, Sutton Bonington Campus, University of Nottingham, Loughborough LE12 5RD, UK; robert.stott@nottingham.ac.uk (R.J.S.); toshana.foster@nottingham.ac.uk (T.L.F.); 4Department of Microbiology and Immunology, University of Nevada, Reno, NV 89557, USA

**Keywords:** vaccine, microvesicles, delivery system, infectious diseases, immunity

## Abstract

The prevention and control of infectious diseases is crucial to the maintenance and protection of social and public healthcare. The global impact of SARS-CoV-2 has demonstrated how outbreaks of emerging and re-emerging infections can lead to pandemics of significant public health and socio-economic burden. Vaccination is one of the most effective approaches to protect against infectious diseases, and to date, multiple vaccines have been successfully used to protect against and eradicate both viral and bacterial pathogens. The main criterion of vaccine efficacy is the induction of specific humoral and cellular immune responses, and it is well established that immunogenicity depends on the type of vaccine as well as the route of delivery. In addition, antigen delivery to immune organs and the site of injection can potentiate efficacy of the vaccine. In light of this, microvesicles have been suggested as potential vehicles for antigen delivery as they can carry various immunogenic molecules including proteins, nucleic acids and polysaccharides directly to target cells. In this review, we focus on the mechanisms of microvesicle biogenesis and the role of microvesicles in infectious diseases. Further, we discuss the application of microvesicles as a novel and effective vaccine delivery system.

## 1. Introduction

Vaccines are one of the most critical healthcare interventions, protecting millions of people around the world and contributing to the decreased incidence of a variety of infectious diseases and associated fatality rates. Vaccines induce cellular and humoral immune responses to infectious pathogens such as bacteria, fungi and viruses with limited side effects [1]. Currently, vaccines protect against more than 25 debilitating and life-threatening diseases, including measles, polio, tetanus, diphtheria, meningitis, influenza, typhoid, and cervical cancer [2]. However, there are still many infectious pathogens for which there are no approved vaccinations [3]. Additionally, research is continually conducted to further improve vaccine efficacy and duration of protection, as well as increase public confidence in vaccine safety [4,5,6].

Classical vaccines, such as live-attenuated vaccines (LAVs) and whole-cell inactivated vaccines (IV), have demonstrated efficacy in controlling infectious disease outbreaks [7]. However, due to limitations, most prominently their contraindication in pregnant and immunocompromised patients [8,9], research is continually necessary to improve the efficacy of these vaccines and develop alternative approaches [10]. One alternative is the use of subunit and nucleic acid-based vaccines. These are suggested to be a safer approach due to limited side effects and reduced chance of reversion to virulence through back-mutation of attenuating mutations in LAVs [11]. Another alternative currently under investigation is the use of novel vehicles for vaccine delivery such as cell-derived, lipid bi-layered extracellular vesicles (EVs), which could improve antigen retention at the site of injection, facilitate delivery to immune cells and increase overall immunogenicity [12,13,14]. EVs play a crucial role in intercellular communication, acting as vehicles for the transport of membrane and cytosolic proteins, lipids, and nucleic acids, including differential RNA molecules [15,16,17,18]. They have also been shown to be potential carriers for therapeutic molecules, including piceatannol [19] and RNA drugs such as antisense oligonucleotides, Cas9 mRNA, guide RNAs [20] and miRNAs [21].

Importantly, EVs could serve as prospective vaccine candidates against bacteria and parasites [22]. Bacteria-derived vesicles are capable of interacting with innate immune cells, e.g., macrophages and neutrophils, as well as adaptive immune cells and antigen-presenting cells (APCs), e.g., dendritic cells (DCs), and thus may lead to protective immune responses [23,24,25,26].

Further, it has been demonstrated that EVs obtained from genetically modified cells exhibited a neuromodulatory effect in autoimmune and neurodegenerative diseases [27,28,29]. In addition, Horrevorts et al. reported that glycan-modified apoptotic melanoma-derived EVs resulted in enhanced priming of tumor-specific CD8+ T cells [30]. Thus, EVs obtained from genetically engineered cells could be also considered as vaccine candidates. 

Indeed, application of EVs in vaccine delivery might be prospective in treatment of several types of cancers; the mechanisms, advantages, and prospects of EVs as antigen-carrier vaccines in cancer vaccine development have been previously extensively reviewed [31,32]. Furthermore, several studies have also reported the potential applications of EVs in vaccines against infectious diseases [33,34]. 

EVs can be differentiated into exosomes, microvesicles (MVs) and apoptotic bodies, depending on their origin, and can also differ in their function for the development of therapeutics. In this review, we focus on one type of EVs: MVs and how their multiple advantages as delivery systems depend on their natural properties such as biocompatibility, enhanced stability, limited immunotoxicity and specific cell targeting properties [35,36]. We also discuss MVs as a potential novel delivery approach of nanoscale vaccines [12].

At present, multiple vaccines have been produced, each with advantages and limitations [1]. LAVs contain the attenuated pathogen, which are modified to decrease virulence in comparison with the wild-type pathogen [37,38]. One advantage of these vaccines is that they can mimic natural infection, inducing a robust immune response and confer immunity lasting for years after a single dose [39]. However, in rare cases, LAVs can undergo back-mutations, reverting the introduced attenuating mutations and thereby leading to replication of the wild-type infectious pathogen. This can therefore lead to the progression of the disease and in very rare cases, death [37,40]. 

Another current vaccination method is the use of whole-cell and subunit inactivated vaccines (IVs) [41,42]. Whole-cell IVs are heterogeneous in antigenic composition and therefore can elicit non-specific immune responses [1]. In contrast, subunit IVs include purified antigenic components, such as proteins and polysaccharides instead of whole microorganisms, which can minimize side effects and still induce an immune response [43]. A downside of IVs is reduced immunogenicity compared with LAVs due to a lower concentration of antigens and the lack of elements enhancing innate immunity [42]. A novel approach for vaccine design includes delivery of nucleic acids such as DNA or RNA, which encode specific target antigens that thereby elicit immune responses in the host [44,45]. These vaccines are gaining popularity due to their relatively simple design requirements, cost-effectiveness and ease of production. However, these vaccines are currently in various stages of clinical trials [45,46,47], and due to the global public health COVID-19 emergency, they have only very recently been approved for human use [48,49,50].

## 2. Vaccine Delivery Systems

The route of vaccine delivery is important for uptake and processing of the target antigen by antigen presenting cells (APCs), which in turn activate immune cells [51,52]. Therefore, the route of administration is essential for vaccine efficacy [51]. Vector-based vaccines (VBVs) combine the advantages of both live and subunit vaccines [45]. In general, VBV constructs contain non-pathogenic viruses as a vector backbone carrying genes, which encode for the antigens of pathogens [53]. The backbone can be derived from live-attenuated or non-replicating vectors [47]. Retroviruses, herpes simplex virus, measles virus, adenoviruses and poxviruses are commonly used [53]. The efficacy of several VBVs has been evaluated in clinical trials (Clinicaltrials.gov, NCT04128059, NCT03333538). For example, a recombinant adenovirus vector has recently been used in the development of a COVID-19 vaccine (Clinicaltrials.gov, NCT04313127, NCT04552366) [54]. However, VBVs have a potential limitation linked to pre-exposure to the virus from which the backbone is derived; this can induce neutralizing antibodies and reduce vaccine immunogenicity [53]. 

Nanoscale vaccines (NVs) have recently become a novel approach for vaccine delivery [55]. These types of vaccines include virus-like particles (VLPs), polymeric, inorganic nanoparticles, liposomes and nano-emulsions and have demonstrated efficacy both in vitro and even *ex vivo* [55,56]. Recently, two lipid nanoparticle-formulated, nucleoside-modified RNA vaccines were developed against SARS-CoV-2 [48,50]. These vaccines, BNT162b1 and BNT162b2, containing mRNA encoding the SARS-CoV-2 receptor-binding domain and membrane-anchored SARS-CoV-2 full-length spike protein, respectively, were tested in clinical studies (Clinicaltrials.gov, NCT04368728, NCT04368728) and were shown to be safe and immunogenic [57]. This demonstrates that NVs can be used as a platform for developing vaccines against emerging infections, given their desired features including prolonged stability, immunogenicity and a non-invasive route of administration [58,59,60]. The future use of NVs as vaccines will, however, require optimization of particle size and polarity, enhanced tissue penetration, and the means to prevent potential immunotoxicity in order to overcome limitations identified in in vivo studies [58,61,62]. 

Novel delivery systems using MVs, however, have multiple advantages in the delivery of biological substances compared with other nanoscale vehicles. Due to their endogenous origin, MVs are less toxic than other synthetic liposomes or polymeric nanoparticles [58]. They are also capable of crossing natural biological barriers including the blood–brain barrier [63,64,65] and remain stable due to their lipid-rich membrane structure, which also ensures resistance to detergents [66]. In addition, the range in size of MVs influences their uptake by antigen presenting cells to elicit specific immune responses against the cargo within the MV [23,67].

## 3. The Biogenesis of MVs

MVs, also known as ectosomes, are nano-sized EVs that are released from cells into the intercellular environment [60]. In contrast to exosomes, which are derived from endosomal compartments, MVs are produced by shedding directly from the plasma membrane as the result of apoptosis or membrane remodeling [68]. They are commonly found in various body fluids such as saliva, milk, urine, blood and serum [60,69,70]. Uptake of MVs by target cells is largely mediated by receptors that interact with the universal molecules present in the MV membrane, such as lipids and specific peptides (Figure 1). MVs range between 100 and 1000 nm in diameter and are therefore easily internalized into target cells by receptor-mediated endocytosis or phagocytosis [60,71]. MVs have been shown to deliver cargo and elicit strong cellular and humoral immune responses [58]. The advantages of MVs as potential vehicles for vaccines could result from their natural origin as they are produced by multiple cell types in physiological and pathological conditions [63]. 

Although MVs originate from different cell types, they can share a similar composition of proteins and lipids and express common host markers, including major histocompatibility complex (MHC), flotillin, and heat shock 70-kDa proteins [69,72]. The content of MVs has also been shown to be enriched with other cellular proteins (tetraspanins, CD9, CD63, CD81 receptors), cytoskeletal proteins (actin, tubulin) and lipids (phosphatidylserine, ceramide and cholesterol) that are indicative of the cells from which they are derived [69,73] (Figure 1). Some proteins, such as the heat shock protein 70 family and tetraspanins, were originally identified as specific markers for exosomes; however, these proteins have also been identified in MVs and apoptotic bodies [74,75]. Thus, MV cargos can include proteins unique to the cell of origin [69], and therefore receptors expressed on MV membranes can define the cell target [76].

In natural conditions, MVs are generated directly from the plasma membrane by outward budding followed by fusion and release of these vesicles to the extracellular space [77]. Several changes in the cell can initiate the budding process, including alterations to the distribution of asymmetric phospholipids by increasing calcium levels in the cytosol [68,78]. Increased concentration of calcium ions activates the calcium-dependent enzymes scramblase and floppase and inhibits the activity of translocase [79,80]. This prevents translocation of phosphatidylserine (PS) and phosphatidylethanolamine (PE) to the inner side of the membrane, thus disturbing the phospholipid composition of the plasma membrane [68]. As a result, MVs bud from the cell membrane with both PS and PE present on the surface [78,81].

Cytoskeletal rearrangement also contributes to the production of MVs. Changes in the cytoskeleton lead to detachment of the plasma membrane from the cortical cytoskeleton, including actin fibers capped by proteins, which maintain cell stability and shape [77,78]. Calpine, a calcium-dependent enzyme, cleaves actin-capping proteins, which disorganizes the cytoskeleton and leads to MV release [68]. Additionally, cytoskeletal changes can be caused by reduced synthesis of phosphatidylinositol 4,5-biphosphate (PIP2), which is essential for the attachment of the plasma membrane to the cytoskeleton [68,82].

MVs can also be produced during apoptosis. Amongst the proteins responsible for MV shedding through apoptosis is Rho [83], a GTPase family protein. Upon activation, Rho proteins exchange GDP for GTP and signal to downstream Rho-associated kinase (ROCK) [84]. GTP-bound Rho thereby activates ROCK. This activation phosphorylates LIM kinase (LIMK), which prevents cofilin from severing actin filaments and prolongs the extension of actin fibers [85]. Activated ROCK also inhibits myosin light chain phosphatase (MYLP) activity, resulting in an enhancement of myosin phosphorylation [86], actin-myosin sliding, detachment of the plasma membrane from the cytoskeleton, and the release of MVs [87].

MVs play an essential role in cellular communication in physiological and pathological conditions, contributing to the pathogenesis of inflammation [88,89,90,91,92,93,94], coagulation [88,95,96,97], immunomodulation [93], regeneration [98,99,100,101] and tumorigenesis [76,102]. MV contribution to pathogenesis is often achieved through delivering cytokines, chemokines, mRNA and miRNA between cells [103,104]. Thus, MVs are considered as potential biomarkers in various diseases [69,105,106].

## 4. MVs as Immune Modulators

MVs represent a critical component of the immune response for their ability to transfer cytokines, interleukins, growth factors, and other biomolecules from cell to cell [106]. Therefore, they can contribute to the control of inflammation and the adaptive immune response [107]. For example, it has been shown that MVs derived from platelets (pMVs) contribute to inflammation by facilitating leukocyte adhesion to endothelial cells [108]. This is due to expression of PS on the surface, along with adhesive glycoproteins such as GP1b (CD42b), GPIIbIIIa (αIIbβ3 integrin; CD41, CD61), β1-integrin (CD29), and P-selectin (CD62P) [109,110,111]. These molecules can immobilize circulating neutrophils, retaining them at the site of inflammation [106]. pMVs are also capable of transporting pro-inflammatory cytokines and chemokines, such as interleukins (IL-1β, IL-8, IL-6, and RANTES), to the site of leukocyte-endothelial cell adhesion and promote inflammation [106,112]. 

Leukocytes, including NK cells (natural killer), have been shown to release MVs [113]. NK-derived MVs (NKMVs) are known to express perforin, granzymes, CD40L and other molecules involved in cytotoxicity, homing, cell adhesion, and immune activation [113,114,115,116]. It was demonstrated that NKMVs from healthy donors can activate peripheral blood mononuclear cells (PBMCs) by activating CD4+ T cells and inducing the expression of co-stimulatory molecules on monocytes and CD25 on T cells [116]. In inflamed lymph nodes, NKMVs are suggested to contribute to NK cell priming of CD4+ T helper (Th) type 1 cells via secretion of interferon-γ (IFN-γ) [117]. NKMVs are also shown to express several tumor necrosis factor (TNF) receptors and ligands and carry IFN-induced transmembrane proteins, suggesting that they can also modulate adaptive immune responses by directly activating T and B cells [116]. The activation potency of immune cell-derived MVs has also been demonstrated in an immune-tolerant and immune-suppressing tumor microenvironment [118]. Even though, NKMVs were exposed to tolerogenic stimuli [119] and an immune-suppressing environment containing IL-10/TGFβ in order to mimic the tumor microenvironment, they were shown to stimulate monocytes and induced T-cell activation [116]. 

Granulocytes, another subset of leukocytes, can also produce MVs [120,121,122]. Shen et al. have demonstrated that MVs released by apoptotic polymorphonuclear neutrophils (apoPMN-MVs) can significantly bind to CD25 (IL-2Rα)- CD127 (IL-7Rα)+ Th cells with higher capacity than apoptotic neutrophils and can selectively suppress proliferation of these lymphocytes [123]. After an acute infection immediately following pathogen elimination, apoPMN-MVs were shown to suppress proliferation of CD25- CD127+ Th cells by downregulating expression of IL-2 and IL-2R, thereby supporting immunological tolerance [124,125]. However, these MVs failed to affect T-cell proliferation, suggesting that these T cells likely produced substantial amounts of IL-2, secreted IL-2 and CD25 [123].

## 5. Mesenchymal Stem Cell (MSC)-Derived MVs

Mesenchymal stem cell (MSC)-derived MVs were introduced as one of the most attractive pharmaceutical carriers due to specific properties inherited from the parental cells [126]. Maumus et al. have reviewed the advantages of MVs derived from MSCs and their therapeutic applications [127]. MSC-derived MVs were shown to simulate the immunoregulatory and regenerative actions of MSCs [128,129]. Thus, MSC-derived MVs lack MHC class I and II, which allows allogenic transfusion without eliciting autoimmunity or tumors [98,130]. MSC-derived MVs were shown to circulate in the blood and contain molecules, including miRNA, with potential therapeutic properties [131]. Collino et al. have also demonstrated that MSC-derived MVs can hold ribonucleoproteins involved in the intracellular trafficking of RNA as well as selected miRNAs [132]. This transfer of miRNAs by MVs to target cells highlights the possibility that the biologic effect of stem cells could depend on MV-shuttled miRNAs [132].

MSC-derived MVs can modulate immune responses via interaction with leukocytes [133,134]. The immunomodulatory properties of MSC-derived MVs were confirmed by Mokarizadeh et al., who showed that the expression of PD-L1, galectin-1 and membrane-bound TGF-β inhibited auto-reactive lymphocyte proliferation and promoted the secretion of anti-inflammatory cytokines IL-10 and TGF-β [135]. Also, Henao Agudelo et al. have demonstrated that MSC-derived MVs can promote a regulatory-like phenotype in M1-macrophages, which showed higher CD206 levels and decreased CCR7 expression [136]. This effect was also associated with reduced levels of inflammatory molecules (IL-1β, IL-6, nitric oxide) and increased expression of immunoregulatory markers (IL-10 and arginase) in M1-macrophages. 

MSC-derived MVs also have therapeutic potential in neuropathology [137,138,139,140]. Jaimes et al. have reported that MSC-derived MVs can prevent TNF-α, IL-1β and IL-6 upregulation in microglia cell lines and primary microglia cells treated with lipopolysaccharides (LPS) [133]. Additionally, a co-culture of microglia cells with MSC-derived MVs could upregulate CCL22 expression and can be used as a marker for M2 microglia phenotypes [133,141] characterized by enhanced phagocytosis and anti-inflammatory properties [142]. MSC-derived MVs were also shown to inhibit the expression of the activation markers CD45 and CD11b on microglia cells treated with LPS, suggesting an anti-inflammatory effect [133].

## 6. MSC-Derived MVs Isolation and Purification Techniques

Prospective immunomodulatory properties of MSC-derived MVs and potential application of MVs as a vehicle for the delivery of target proteins have initiated the development of various techniques for their large-scale manufacture (Figure 2) [143]. However, lack of bioprocessing methods for scaling up the derivation of MSC-derived MVs has become a serious challenge both for research and clinical purposes. Several studies have demonstrated that MSC 3D-cultures are more beneficial than monolayer cultures in several therapeutic applications [144,145]. Previously, Bartosh and Frith have reported that 3D MSC-assembly preserves the phenotype and innate properties of MSCs and promotes the microenvironment similar to that in vivo [146,147]. Thus, it has been suggested that 3D MSC-assembly could be used for the production of therapeutic MVs [126,148]. Using the unique composition of a Polyethylene glycol (PEG) hydrogel microwell-array, Cha et al. have demonstrated that 3D MSC-assembly increases MV production up to 100-fold in comparison with 2D cultures [126]. 

Another approach includes treatment of cells with cytochalasin B, which disrupts the bonds between actin fibers leading to disintegration of the cytoskeleton [143]. Since various sized vesicles are produced, MVs are differentiated from exosomes by using multiple differential centrifugation steps [150]. Due to the cytochalasin B-induced ability of MVs to fuse with cell receptors in a similar way to naturally produced MVs, it was suggested that they also have similar physiological and biological characteristics, including the transport of biologically active molecules between cells [143]. 

Ultracentrifugation is one of the most reliable and validated techniques for the robust isolation and purification of large-scale MVs [149]. Using this approach, vesicles can be isolated from cells and, due to size variation, differential centrifugation can separate exosomes from MVs [149]. 

Aside from differential centrifugation, various approaches have been described to separate MVs from exosomes [151,152,153]. Density gradient centrifugation remains one of the most utilized techniques [154], in which EVs can be separated into specific layers based on their buoyant density in iodixanol solutions [154]. Additionally, by means of size-exclusion chromatography using porous beads, Boing et al. have separated EVs based on their hydrodynamic radius [155]. Immune-affinity capture utilizing monoclonal antibodies against surface proteins is also often adopted for the isolation of EVs, particularly using antibodies against tetraspanins present in exosomes [156]. Recently, several studies have shown that all EVs express tetraspanins at different levels, so the purification methodology yields a mixed population of different EVs [74,75]. However, immune-affinity methods require clear data on the biomarkers that distinguish MVs from exosomes, and at present, these have not been clearly defined [152]. The size of the EV sample in represented purification methods is limited by different variables, including the size of the centrifuge and rotary tubes used, the size of the column, and the quantity of antibody coated beads used [153,157,158]. Thereby, a suitable method of isolation and purification of MVs needs to be applied for large-scale production and further therapeutic delivery to target cells.

## 7. Tumor-Derived MVs

Tumor-derived MVs have been shown to contribute substantially to cancer progression [102,159]. These MVs can stimulate cancer cell migration, invasion of tissues and proliferation [160]. Features of these MVs that support tumor growth and invasiveness are linked to interactions with the tumor microenvironment, where they can induce a tumor-supporting milieu [161,162]. Bordeleau et al. have suggested that tumor-derived MVs could contribute to extracellular matrix reorganization by enhancing the contractility of non-malignant epithelial cells within the primary tumor and the metastatic sites [163]. Also, Taheri et al. have demonstrated that glioma tumor-derived MVs can stimulate proliferative and metastatic gene expression in normal astrocytes within the tumor microenvironment, thus affecting tumor growth and invasion [164].

Another mechanism utilized by tumor-derived MVs to support malignancy is the reprogramming of immune cells. It was shown that these MVs can modify leukocyte differentiation into tumor-supporting cells as they stimulate growth, migration and tube formation of tumor cells [162,165]. For example, Ma et al. have reported that tumor-derived MVs stimulate the conversion of macrophages into an active phenotype that expresses cytokines and growth factors, leading to tumor growth and the formation of metastases, as well as differentiation of tumor stem cells [166]. In addition, tumor-derived MVs enable cancer cells to escape immune surveillance by suppressing NKs and CD8+ T lymphocytes [167,168,169]. This immunosuppressive effect was shown to be associated with the secretion of immunosuppressive cytokines by regulatory B cells [170]. 

Conversely, tumor-derived MVs have been shown to contribute to the anti-cancer immune response. MVs, when engulfed by dendritic cells, could contribute to antigen-processing, thus promoting the selection of tumor antigens with immunogenic properties [171]. Pineda et al. have demonstrated that irradiated tumor-derived MVs could facilitate the capture of tumor-associated antigens by antigen-presenting cells, thereby triggering an innate and adaptive immune response against malignancy [172]. Interestingly, the nature of tumor-derived MVs, as they form particles, was shown to make them more immunogenic than the soluble tumor antigens [173]. This was confirmed by Rughetti et al., who showed that the cell surface-associated tumor antigen Mucin 1 (MUC1) was immunogenic only when cross-processed and presented to antigen-specific CD8+ T cells carried by MVs [174].

It appears that the efficacy of tumor-derived MVs in eliciting an anti-tumor immune response makes them an attractive potential vaccine platform against cancer. However, their potential immunosuppressive activity severely limits this clinical application.

## 8. Role of MVs in Infectious Diseases

MVs play an important role in defense against infectious diseases. For example, it was shown that pro-coagulant MVs, isolated from M1 protein-stimulated PBMCs, can limit the dissemination and growth of bacteria [175]. Also, Oehmcke et al. have shown that formation of the plasma thrombus resulted from association between pro-coagulant MVs and Streptococcus pyogenes [176]. Additionally, it was reported that the MVs containing parasite RNAs from malaria-infected erythrocytes could fuse with NK cells and activate them to provide the first line of defense against parasite infection [177]. The effect of MVs in the pathogenesis of coagulopathy was demonstrated in meningococcemia, Ebola hemorrhagic fever [178] and Chaga’s disease [179].

MVs can affect the function of macrophages, which are essential to the clearance of microbial pathogens [180]. MV delivery of pore-forming toxins to macrophages can induce their polarization into a novel CD14+MHCIIlowCD86low phenotype, which is characterized by an enhanced reactivity to Gram-positive bacterial ligands [181]. It was reported that MSC-derived MVs have potential therapeutic activity in mouse models of bacterial pneumonia [134,182,183]. Monsel et al. showed several potential mechanisms underlying the protective effect of these MVs. Firstly, there was an increased clearance of bacteria from the lungs, which was explained by stimulation of monocyte phagocytosis [134]. Secondly, MVs could affect monocyte and alveolar macrophage activities by suppressing cytokine-induced lung injury and lung protein permeability [134]. Another suggested mechanism includes the effect on metabolism of alveolar epithelial type 2 cells, where MVs carrying metabolic enzymes could facilitate microbial clearance [134].

In short, MVs serve as messengers between the host and the pathogen and are also used for defense. It is very important that pathogen-derived MVs carrying pathogen particles could elicit a host immune response by activating immune cells including monocytes, macrophages, T cells and NK cells [184,185,186]. Thus, MVs could be considered as a potential vaccine candidate against infectious diseases. 

## 9. MVs in Vaccine Applications against Infectious Diseases

Recent studies have demonstrated the potency of MVs as a vaccine delivery system, using MVs derived from bacteria to deliver immunogenic epitopes [12,187]. For example, bacterial outer membrane vesicles (MVs), naturally produced by Gram-negative bacteria, were shown to have nano-sized lipid-bilayer vesicular structures composed of multiple immunostimulatory components (Figure 3) [187,188]. MVs derived from Neisseria meningitides and Vibrio cholera have been incorporated into licensed vaccine formulations [189,190,191]. However, it should be noted that non-commensal, pathogen-derived MVs as vaccine delivery systems currently have several limitations including unintended toxicity, low expression of some heterologous antigens, and variable efficacy depending on source and formulation [187]. Bioengineering bacteria-derived MVs has been addressed in several studies with the aim that MV-based vaccines could have more therapeutic applications and overcome these limitations [67,192,193]. For example, LPS is a potent immune system activator, but it can induce severe immunotoxicity [194]. Some techniques that depend on detergent treatment of bacteria have been used to achieve LPS detoxification, and therefore production of MV vaccines lack LPS, such as the current meningococcal vaccine [195]. Otherwise, Zariri et al. have genetically detoxified LPS with potent activation of the innate immune system by TLR4 [196]. Additionally, Price et al. have developed a bioengineered bacteria-derived MV-expressing heterologous glycan antigen instead of a protein antigen to elicit appropriate immunity against the pathogen of interest [197]. Alternatively, commensal non-pathogenic bacteria-derived MVs could be used to reduce toxicity and improve safety. Non-pathogenic Gram-negative bacteria have been engineered to target specific antigens either in the MV lumen or on the MV surface (Figure 3) [12,198,199]. 

The efficacy of antigen delivery by MVs could be improved by presenting antigens on their surface, and there are several approaches that could ensure antigen localization on the surface. One method includes displaying the target protein on the membrane by fusion with surface anchor heterologous proteins [197,200]. This was demonstrated by Rappazzo et al. when a ClyA surface protein of MVs was fused with the ectodomain of influenza A matrix protein 2 [201]. This vaccine containing tandem heterologous M2e peptides (M2e4xHetOMV) ensured 100% survival against lethal doses of the mouse-adapted H1N1 influenza strain PR8 by triggering TLR1/2, TLR4, and TLR5. Moreover, passive transfer of antibodies from M2e4xHet-OMV vaccinated mice to unvaccinated ones also resulted in 100% survival [201].

In another approach, target proteins were displayed on the surface of MVs by over-expression of the outer-membrane proteins of the pathogen. For example, it was shown that over-expression of the N. meningitidis outer-membrane recombinant protein, NspA, increased its packaging into MVs [202]. Also, expression of the target antigen as a fusion with periplasmic signal proteins was shown as an alternative approach to target antigens on the surface of MVs [198]. This method was demonstrated by Muralinath et al. when a MV-derived vaccine was developed by expressing the PspA protein of S. pneumoniae fused with a periplasmic signal protein [203]. 

MVs have many advantages compared with both live and inactivated vaccines [12,187]. MVs can be administered in non-invasive ways (orally or nasally, without using needles), thus making a booster dose possible and enabling mass vaccination programs in challenging environments and at relatively low cost [12]. Moreover, MV-based vaccines can specifically target mucosal sites, which are often not targeted by other injectable vaccines [12]. Importantly, they can also induce innate and adaptive immune responses [12,204]. The safety of these vaccines is based on their non-cellular form and the absence of infectious components [12]. One of the indisputable advantages of MV-based vaccines is their stability in liquid and frozen forms [205]. Furthermore, MV-based vaccines, especially bacteria-derived MVs, have been considered as an emergency vaccine for arresting the spread of future epidemics [206,207]. These MVs can be rapidly manufactured and modified as well as quickly up-scaled with low technological complexity, high flexibility for producing a wide range of vaccines at low cost and potential thermo-stability of the formulated product [12,205,207,208,209]. 

## 10. Potential Clinical Applications of MVs

Given that MVs can serve as biomarkers for various diseases and can often contribute to their pathogenesis, they therefore have the potential to be diagnostic and therapeutic tools [210,211,212,213]. It has been suggested that MVs could be used as biomarkers for the progression of neuropathologies and malignancies [214]. Mege et al. have demonstrated the use of tumor-derived MVs to monitor colon and rectal cancer progression [215]. In another study, Colombo et al. demonstrated that MVs in cerebrospinal fluid could be used to monitor infections of the nervous system and the progression of multiple sclerosis [94]. MVs could also serve as markers of regeneration and repair, as demonstrated in cases of acute kidney injury and liver fibrosis [98,100,216,217].

The ability of MVs to deliver cargo is an attractive model for the delivery of therapeutics to sites commonly hard to access via conventional delivery routes [58,104]. For example, Tang et al. have assessed the delivery potential of tumor-derived MVs when used to transport chemotherapeutic drugs [66]. In addition, Usman et al. have reported that red blood cell-derived MVs can carry RNA drugs, including antisense oligonucleotides, Cas9 mRNA and guide RNAs [20]. RNA drug delivery with these MVs possessed highly robust microRNA inhibition and CRISPR–Cas9 genome editing in both human cells and xenograft mouse models [20]. Additionally, Wang et al. have demonstrated that MVs can be used to upload and transport p53, an anti-tumor therapeutic protein [58]. Although MVs were shown to possess a high potency for both vaccine and drug delivery, they still face challenges concerning storage conditions, low yield production, quality control and targeted delivery [218,219].

## 11. Conclusions

We have highlighted the potential of MVs as key carriers of molecular information and their role in providing functional information on physiological and pathological processes. Depending on the natural and unique properties related to their origin, MVs have been used as delivery vehicles toward specific cells and tissues. Here, we reviewed the role of MVs in therapeutics against infectious diseases and highlighted the potential of MVs as antigen-carrying systems that can improve vaccine efficacy and overcome potential classical vaccine limitations. Biological safety and low-cost production are indisputable advantages of this approach to vaccine delivery. Additionally, this method expands on the excellent opportunity for immunogenic cargo delivery on a large scale. The possible modification of MVs with specific antigens to induce innate and adaptive immunity is an attractive feature of MVs. Indeed, MV-induced immune responses could be more robust than classical vaccine delivery methods since MVs protect their contents from degradation when delivering cargo to target cells. Still, there are several limitations in the application of MV-based vaccines, including low yields under natural conditions. We have discussed some techniques to improve MV production, such as MSCs. We highlighted that MVs can be used as future tools for diagnosis, treatment, and prophylaxis against several diseases. Therefore, the development of novel approaches for standard isolation, purification, and mass production remains important for future applications of MVs as vaccine delivery and therapeutic vehicles.

## Figures and Tables

**Figure 1 ijms-22-01142-f001:**
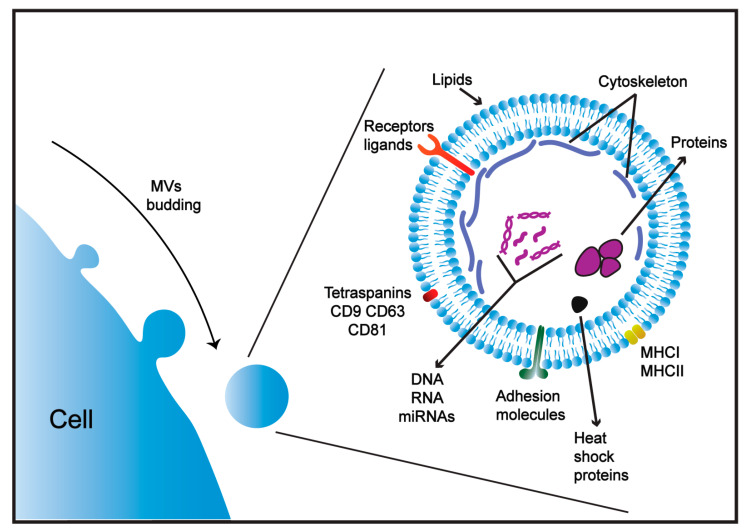
Structure of microvesicles (MVs). MVs consist of a phospholipid membrane bi-layer and contain transmembrane proteins (ligands and receptors), adhesion molecules and common markers including major histocompatibility complexes (MHC), and heat shock 70-kDa proteins. MVs are enriched with tetraspanins and cytoskeletal proteins and possess a set of proteins and nucleic acids unique to the cell of origin.

**Figure 2 ijms-22-01142-f002:**
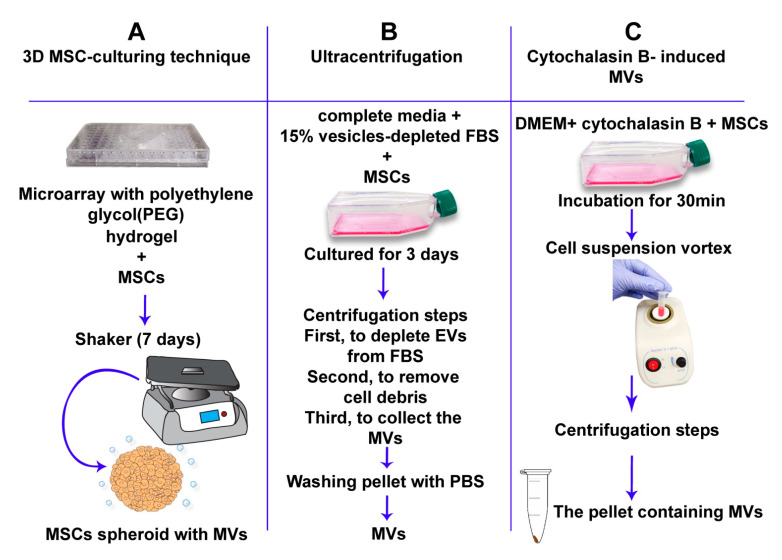
Formation of artificial MVs. Three separate approaches for the artificial formation of MVs. (**A**) The 3D mesenchymal stem cell (MSC)-culturing technique uses polyethylene glycol hydrogels to form spheroid MSCs. MSC-spheroids are subsequently cultured for 7 days on a 30-rpm orbital-shaker (3D w/shaking). Scalable pro-duction of MSC-derived MVs can be achieved by this method. (**B**) The ultracentrifugation approach. This includes numerous centrifugation steps. MSCs are first cultured with complete media supplemented with 15% vesicle-depleted fetal bovine serum (FBS) for 3 days. MSCs are then centrifuged at 120,000× g for 18 h to deplete extra-cellular vesicles from FBS. The cell solution is centrifuged at 300× g for 30 min at 4 °C to remove whole cells and large debris. Supernatants are then centrifuged at 16,500× g for 20 min at 4 °C to collect the MV fraction [149]. (C) Cytochalasin B-induced MVs. This approach depends on the incubation of MSCs with cytochalasin-B to obtain MVs.

**Figure 3 ijms-22-01142-f003:**
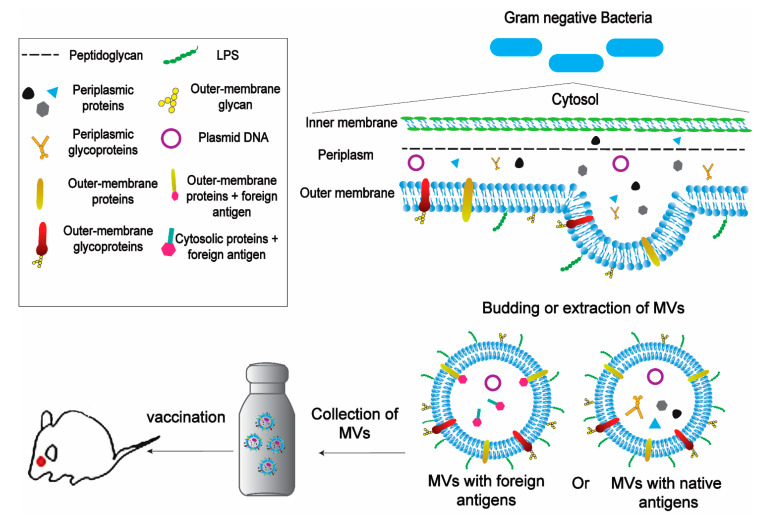
Schematic representation of the application of bacteria-derived MVs in vaccines. MVs can be obtained from Gram-Negative Bacteria naturally by budding or by detergent extraction. MVs can carry native antigens (including proteins, glycans and glycoproteins) or can be engineered to carry foreign antigens fused with outer-membrane proteins or cytosolic proteins. The next step is to isolate and purificate MVs and then collect them in large scale amounts.The MVs’ amount has to be suitable for using in the vaccination process. Then, the MV-based vaccine is performed on animals in non-invasive ways with the possibility of a booster dose.

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
