# Peer review of "Membrane Microvesicles as Potential Vaccine Candidates"

_ijms, 2021, doi:10.3390/ijms22031142_

Round 1
Reviewer 1 Report
This review topic of using membrane microvesicles (MVs) as vaccine candidates fits well with the current SARS-CoV-2 global epidemics. The manuscript is written in a well-organized manner and quite straightforward. This reviewer only has some minor points, as listed below.
1) What are the challenging aspects and demerits of using MVs as vaccine candidates?
2) Several perspectives of MVs as candidates in clinical applications.
3) Technology establishment of antigen-displaying MVs for certain pathogens.
Author Response
- Undoubtedly, MVs represent an attractive approach in vaccine design due to their low-cost production, immunogenicity and biologicsal safety compared to viral vectors. However, large scale production, optimization of storage conditions and targeted delivery and quality control of MVs might become a serious challenge.
-
We provided information about potential application of MVs in clinic in section 9.
-
Technical aspects of MVs production of certain pathogens are highlighted in section 8. Engineering of outer membrane or surface membrane proteins for MVs production is beneficial in eliciting mucosal and systemic antigen-specific immune responses depending on pathogen.
Reviewer 2 Report
As a reviewer, I enjoyed reading this article. It is well-written logically, and the authors summarized the mechanisms of microvesicle bio-genesis and the role of microvesicles in infectious diseases. Moreover, they described well the application of microvesicles as a novel and effective vaccine delivery system.
I can suggest just a few minor points to improve this paper.
Major concerns:
- The authors should add some descriptions about the role of MVs in cancers. In addition, it would be better to describe the future perspective of MVs in clinical applications to cancer treatment (transporter of cancer vaccine).
- As to the vaccine delivery system, please describe the transporter of mRNA vaccine against COVID-19, since it is the main concern worldwide.
- There are too many hyphens in a single word, such as fa-tality, de-bilitating, and deliver-ing. Please correct these grammatical errors.
Author Response
- Thank you for the remarks. Actually the main focus of the review was the application of MVs as vaccines against infectious diseases. But anyway, we expanded the review and also added section Tumor derived MVs.
- We have provided the description of mRNA vaccine against COVID-19 in Vaccine delivery methods.
- We apologize for these errors that probably occurred during the page-proofs. We have made corrections in the manuscript.
Reviewer 3 Report
The manuscript by Shkair et al offers a review on the use of extracellular vesicles (i.e., microvesicles) for vaccine delivery. Given the current CoVID-19 pandemic, it is opportune to explore novel vaccine and biotherapeutic platforms for the use of emergent or re-emergent pathogens. This manuscript is very well written and offers a great contribution in the field of vaccinology. Although the authors provide relevant information related to microvesicles generation, there is a fundamental gap in the definition of extracelluar vesicles or membrane vesicles and microvesicles. A lot of the protein markers described as cargo in MVs overlap with other extracellular vesicles such as exosomes. This needs to be clearly indicated. For instance, cellular proteins CD9, CD63, CD81 and flotillin-1 are proteins that are found in exosomes. In fact, there are commercially available kits that pull down exosomes from extracellular vesicles preparations using conjugated antibodies against the aforementioned protein markers.
Figure 1: there is a typo on the adhesion molecules. Also please insert a new arrow to indicate “proteins”. The way is currently illustrated it may suggest that there are DNA or RNA binding proteins. Also, another nucleic acids of importance in these vesicles are microRNAs. This is important because later in the text microRNAs are mentioned as MV cargo.
Line 54: the statement “Another alternative currently under investigation is the use of novel vehicles for vaccine delivery such as cell-derived extracellular vesicles (EVs),..” this is the first time that the authors introduce the concept of EVs, however, the tittle suggest that these EVs are considered vaccine candidates and not novel vehicles for delivery. The authors should make the distinction between a vaccine candidate and a method for vaccine delivery.
Line 261: change “galecin-1” to “galectin-1”
Line 293: this whole paragraph should have its own section and should provide more specifics on the different methods of microvesicles purification (methods other than ultracentrifugation had been described). This is particularly important given the overlap-in composition- with other extracellular vesicles such as exosomes. Additionally, use of cytochalasin B increases the yield of all types of membrane vesicles. The authors should not confuse membrane vesicles (MVs) vs microvesicles (MVs). Make sure these differences are well defined.
Line 442: given the current need for rapid manufacturing of vaccines, it would be suitable for the authors to expand on the specifics for this technology and its contribution in rapid manufacturing.
Figure 3: remove hyphens in “anti-gens” “vacci-nation”; “possi-bility”; remove double parenthesis.
Author Response
- We have modified Figure 1 according to your recommendations and replaced the previous one in the body of the paper.
- Thank you for the recommendation. We definitely agree with you that it is necessary to clarify the role of EVs as carriers of various pharmaceutics and vaccine candidates as well. We addressed your comment and have introduced the changes in the paper (labelled with yellow)
- Corrected to "galectin-1"
- We expanded the techniques of microvesicles purification.
- Changes were made.
- Hyphens were removed.
Round 2
Reviewer 2 Report
The authors answered all of my concerns, and the paper was nicely revised.
Author Response
We have made all necessary corrections according to your recommendations.